# Adaptive Sparse Softmax: An Effective and Efficient Softmax Variant for Text Classification

## Abstract

Softmax with the cross entropy loss is the standard configuration for current neural text classification models. The gold score for a target class is supposed to be 1, but it is never reachable under the softmax schema. Such a problem makes the training process continue forever and leads to overfitting. Moreover, the "target-approach-1" training goal forces the model to continuously learn all samples, leading to a waste of time in handling some samples which have already been classified correctly with high confidence, while the test goal simply requires the target class of each sample to hold the maximum score. To solve the above weaknesses, we propose the **A**daptive **S**parse softmax (AS-Softmax) which designs a reasonable and test-matching transformation on top of softmax. For more purposeful learning, we discard the classes with far smaller scores compared with the actual class during training. Then the model could focus on learning to distinguish the target class from its strong opponents, which is also the great challenge in test. In addition, since the training losses of easy samples will gradually drop to 0 in AS-Softmax, we develop an adaptive gradient accumulation strategy based on the masked sample ratio to speed up training. We verify proposed AS-Softmax on a variety of multi-class, multi-label and token classification tasks with class sizes ranging from 5 to 5000+. The results show that AS-Softmax consistently outperforms softmax and its variants, and the loss of AS-Softmax is remarkably correlated with classification performance in validation. Furthermore, adaptive gradient accumulation strategy can bring about $1.2\times$ training speedup comparing with the standard softmax while maintaining classification effectiveness.

## 1 Introduction

Softmax is widely used as the last-layer activation function of a neural classification model. It normalizes the output of a network to a probability distribution over predicted output classes. Softmax is extremely appealing in both academics and industries as it is simple to evaluate and differentiate. Many research efforts are devoted to refining softmax. On one hand, several works (Shim et al., 2017; Liu et al., 2017) have explored accelerating the training process by outputting a subset of classes to reduce computational complexity. On the other hand, improving training effectiveness is also an important direction. For example, Liu et al. (2016) and Liang et al. (2017) enlarged the margin of intra-class and inter-class in softmax, while Szegedy et al. (2016) used label smoothing to prevent the neural networks from excessively trusting gold labels.

Despite the probability form, each output score of softmax is within the range of $(0, 1)$, unlike the sparse probability distribution in real classification tasks. Consequently, the weights of all the classes have to be updated endlessly, which wastes a lot of time and leads to overfitting (Sun et al., 2021). In addition, the training goal of softmax with the cross entropy loss is to make the target score approach to 1, while in test we expect the target score could be superior to scores of other classes. Thus there is an obvious gap between the two objectives. Take the following two prediction cases as an example: It is a 5-class classification task and Class 1 is the gold label.

**Case A:** $\{\textbf{0.4}, 0.15, 0.15, 0.15, 0.15\}$

**Case B:** $\{\textbf{0.49}, 0.5, 0.004, 0.003, 0.003\}$

We can find that in Case A, the score of Class 1 is far larger than all the other classes, leading a successful prediction. By contrast, there is a strong negative class in Case B and its classification result is wrong. Unfortunately, in the training period, since the target score in Case B is much higher than that in Case A, the corresponding cross entropy loss would make the model biased towards improving Case A, although Case B is a hard sample in practice. Our experiments show that sometimes the correlation coefficient between the cross entropy loss of softmax and the classification accuracy is even close to 0 in validation (refer to Table 4).

To address the above-mentioned deficiency, we propose the **A**daptive **S**pare softmax (AS-Softmax) which involves a reasonable and test-matching transformation on top of softmax. Specifically, when computing the softmax activation function, we exclude the classes whose scores are smaller than the actual class to a given margin. In this way, the model focuses on learning to distinguish the target class from its strong opponents, matching the use in test. It is also noted that the loss of AS-Softmax can drop to 0 as long as the score of the target class is much larger than all other classes. In other words, with the progress of training, more and more easy samples will be dropped from back propagation. Thus the learning process tends to find the useful hard training instances, which makes learning more effective. In practice, it is not always the case that increasing the training sample helps the final performance of the model. Yao et al. (2022) and Wang et al. (2017) claimed that training the model with representative instances is more efficient. Easy samples can be discarded by the cascade structure at early stages (Yang et al., 2019) for learning better classification models. Moreover, the increasing removed samples also guide us in developing a novel adaptive gradient accumulation strategy to make the training process faster and faster.

We test AS-Softmax on a variety of text classification tasks, including multi-class classification Socher et al. (2013); Larson et al. (2019), multi-label classification Chalkidis et al. (2021); Kowsari et al. (2017) and token classification Sang & De Meulder (2003); Tseng et al. (2015). All the benchmarks are publicly available and the class sizes range from 5 to 5000+. Experiments show that AS-Softmax consistently improves the performance of text classification by 1% to 2% and alleviates overfitting to a large extent. In addition, AS-Softmax with the adaptive gradient accumulation strategy can bring about $1.2\times$ training speedup while maintaining the performance.

To conclude, AS-Softmax solves the endless training and train-test objective mismatching problems of the original softmax function. Its advantages can be summarized as follows:

- With AS-Softmax, easy samples will be dropped from back propagation gradually , moderating overfitting and making training more effective.
- The loss of AS-Softmax is highly correlated with the final classification performance.
- Its training can be accelerated via the proposed adaptive gradient accumulation strategy.
- AS-Softmax is fairly easy to implement. We have made it as an standard module in Pytorch.[1]

## 2 RELATED WORK

Softmax loss function is widely used as an output activation function for modeling categorical probability distributions. There are a lot of variants of softmax proposed in the literature (De Brebisson & Vincent, 2015; Martins & Astudillo, 2016; Yang et al., 2017; Kanai et al., 2018; Gimpel & Smith, 2010). We will briefly introduce them in terms of efficiency and effectiveness.

**Enhancing Efficiency** Training a model using the full softmax loss becomes prohibitively expensive in the settings where a large number of classes are involved. One important kind of efficient training is to reduce its output dimension to reduce computational complexity. For example, hierarchical softmax (Morin & Bengio, 2005) partitioned the classes into a tree based on class similarities. Meanwhile, many works explored efficient subsets instead of all output classes. We define these methods as *sparse softmax family* uniformly. Jean et al. (2014); Rawat et al. (2019) and Blanc & Rendle (2018) preserves a small number of negative classes in training by importance sampling and kernel based sampling, respectively. Martins & Astudillo (2016) suggested a new softmax variant

---

[1]Code will be released in the final version.

which could generate sparse posterior distributions. Some studies also tried to use approximate estimation to simplify the calculation of softmax. Devlin et al. (2014) and Vaswani et al. (2013) used self-normalization mechanism and noise contrastive estimation separately to simplified the complexity of calculation. Spherical softmax (Vincent et al., 2014; De Brebisson & Vincent, 2015) adopted a quadratic function instead of the original the exponential function, making its parameter update independent of the output size while Shim et al. (2017) introduced the singular value decomposition method to restrict the probability distribution.

**Boosting Effectiveness** Softmax usually lacks the accurate discrimination of similar output classes. Thus, an intuitive idea is to enlarge the inter-class margin and compress the intra-class distribution. Liang et al. (2017) designed a soft distant margin while Liu et al. (2016) incorporated angular margin to obtain a more discriminative decision boundary. The features learned by softmax loss have intrinsic angular distribution Liu et al. (2017). Follow-up studies (Deng et al., 2019; Choi et al., 2020; Wang et al., 2018d; Ranjan et al., 2017; Liu et al., 2017; Wang et al., 2018c) made more effort on the angular constraints and normalization. Additionally, variants in the sparse softmax family like Chen et al. (2021) and Sun et al. (2021) are also in favor of improving the effectiveness of models as they reduce the risk of overfitting. Besides, to prevent the network from becoming over-confident in the current labels, Szegedy et al. (2016) introduced label smoothing while Huang et al. (2021) proposed Noise-Aware-with-Filter to distinguish hard samples from noisy labels. Moreover, Lee et al. (2018) overlay binary masking variables over class output probabilities by dropout to input-adaptively learned via variational inference.

## 3 METHODS

### 3.1 BACKGROUND OF SOFTMAX

Suppose the final output of the neural classifier is $o \in \mathbb{R}^n$. Here $n$ denotes the number of classes. The prediction score of a class $p_i$ in the original softmax is computed as follows:

$$p_i = \text{softmax}(o_i) = \frac{e^{o_i}}{\sum_{j=1}^n e^{o_j}}. \tag{1}$$

Classifiers usually adopt cross entropy combined with softmax as their loss function. For multi-class classification, there is only one actual target class $t$. Then the expression of the loss function and its backward propagation are as follows:

$$\mathcal{L} = -\log(p_t) \tag{2}$$

$$= \log(\sum_{j=1}^n e^{o_j}) - o_t,$$

$$\nabla \mathcal{L} = \frac{\partial \mathcal{L}}{\partial o_j} = \begin{cases} p_j - 1, & \text{if } j = t, \\ p_j, & \text{otherwise.} \end{cases} \tag{3}$$

The cross entropy loss is composed of exponential and logarithmic functions, which is convenient for computation of the loss function and its back propagation. However, as mentioned in Section 1, softmax is trapped in the endless training and train-test goal mismatching problems. In addition, the smaller the cross entropy loss, the stricter the constraints between the target and non-target classes. Sun et al. (2021) proved that in order to make sure the loss $\mathcal{L}$ can be reduced to $\log 2$, the output should satisfy the following inequality:

$$o_t - o_{min} \geq \log(n - 1), \tag{4}$$

where $o_t$, $o_{min}$ respectively means the target (maximal) logit and the minimal logit. When dealing with the high-dimensional classification problems, $\log n - 1$ is a relatively massive but unnecessary margin, which increases the overfitting risk.

In order to mitigate disadvantages aforementioned, we propose an alternative variant of softmax named **A**daptive **S**parse softmax (AS-Softmax).

## 3.2 ADAPTIVE SPARSE SOFTMAX

The training goal of softmax is to make the score of the target class as large as possible. This practice does not match the test objective to encourage the target probability greater than that of any other category. Motivated by Sparse-softmax (Sun et al., 2021), we propose a new training objective which encourages the probability of target class $p_t$ to exceed the probability of non-target class $p_{i \neq t}$ by a specific margin $\delta$:

$$p_t - p_{i \neq t} \geq \delta, \tag{5}$$

where $\delta \in [0, 1]$ is a hyper-parameter.

To achieve this goal, we discard the classes which have already satisfied Eq. 5 in training. Specifically, we adopt a binary factor $z_i$:

$$z_i = \begin{cases} 0, & \text{if Eq. 5 is satisfied and } i \neq t, \\ 1, & \text{otherwise.} \end{cases} \tag{6}$$

Then the modified probability of a class $\tilde{p}_i$ is computed as follows:

$$\tilde{p}_i = \text{AS-Softmax}(o_i) = \frac{z_i e^{o_i}}{\sum_{j=1}^{n} z_j e^{o_j}}. \tag{7}$$

It can be seen that AS-Softmax is extremely easy to implement. Given the output of softmax, AS-Softmax only needs a simple linear screening step while the back propagation process remains the same. With the import of $z_i$, we find that the losses of more and more training samples will reduce to zero, that is to say, these samples are masked, as shown in Figure 1. Therefore, the classification model with AS-Softmax tends to discard easy samples and focus on the rest hard cases, making learning more effective.

In the mathematical forms, AS-Softmax is close to Sparse-Softmax which preserves the fixed top $k$ negative classes to speed up training. However, the loss of Sparse-Softmax is still always larger than 0 and it does not contrast the representations of positive and negative classes.

We also examine a more direct training criterion expressed as:

$$\log(p_t) - \log(p_{i \neq t}) = o_t - o_{i \neq t} \geq \delta'. \tag{8}$$

Nevertheless, $o$ can take any value, namely no upper bound of $\delta'$, which makes it hard to find the optimal value. More comparisons of the above two strategies are provided in Appendix A.1.

### 3.2.1 ALGORITHM DISCUSSION

We can prove that our training object enables the output $o$ to obey the following condition:

$$o_t - o_{min} \geq \log(n\delta + 1). \tag{9}$$

*Proof.* From our criterion in Eq. 5, we can get:

$$p_t - p_{min} = \frac{e^{o_t}}{\sum_{j=1}^{n} e^{o_j}} - \frac{e^{o_{min}}}{\sum_{j=1}^{n} e^{o_j}} \geq \delta. \tag{10}$$

This formula can be further converted into:

$$e^{o_t} - e^{o_{min}} \geq \delta \sum_{j=1}^{n} e^{o_j} \geq n\delta e^{o_{min}}. \tag{11}$$

Namely:

$$e^{o_t} \geq (n\delta + 1)e^{o_{min}}. \tag{12}$$

With a $\log$ transformation, we finally conclude that:

$$o_t - o_{min} \geq \log(n\delta + 1). \tag{13}$$

$\square$

Recall the requirement of softmax in Eq. 4, the criterion of AS-Softmax is much easier to meet, as $\delta$ is quite small in practice.

### 3.2.2 EXTENSION TO MULTI-LABEL CLASSIFICATION

Softmax is typically used in multi-class classification and token classification tasks. However, Su (2020) demonstrated that softmax could also be applied to multi-label classification tasks with the following loss function:

$$\mathcal{L} = \log(1 + \sum_{i \in \Omega_{neg}} e^{o_i}) + \log(1 + \sum_{t \in \Omega_{pos}} e^{-o_t}), \tag{14}$$

where $\Omega_{pos}$ stands for the set of target classes while $\Omega_{neg}$ denotes the set of non-target classes. The training objective of this method is to raise scores of target classes to exceed zero and decrease scores of non-target classes to be lower than zero. Thus in test, the classes with the scores larger than 0 can be regarded as the prediction output. Compared with the sigmoid based method, this method can mitigate the sparse problem when the number of entire output classes is greatly larger than the number of target classes.

Likewise, we extend AS-Softmax to multi-label classification, aiming to encourage probabilities of entire target classes to exceed probabilities of non-target classes by a specific margin $\delta$. To formulate, our additional goal is:

$$p_t^{min} - p_i \geq \delta, \tag{15}$$
$$p_t - p_i^{max} \geq \delta, \tag{16}$$

where $p_t^{min}$ represents the smallest probability of all target classes and $p_i^{max}$ means the largest probability of all non-target classes. To achieve this goal, we define the following binary factors $z_i$ and $z_t$ for multi-label classification tasks:

$$z_i = \begin{cases} 0, & \text{if Eq. 15 is satisfied,} \\ 1, & \text{otherwise,} \end{cases}, z_t = \begin{cases} 0, & \text{if Eq. 16 is satisfied,} \\ 1, & \text{otherwise.} \end{cases} \tag{17}$$

Consequently, the final loss function combined with AS-Softmax can be obtained as follows:

$$\mathcal{L} = \log(1 + \sum_{i \in \Omega_{neg}} z_i e^{o_i}) + \log(1 + \sum_{t \in \Omega_{pos}} z_t e^{-o_t}). \tag{18}$$

Note, if $p_t^{min} - p_i^{max} \geq \delta$ is satisfied, the whole loss will reduce to 0, namely the sample masked.

### 3.3 ADAPTIVE GRADIENT ACCUMULATION STRATEGY

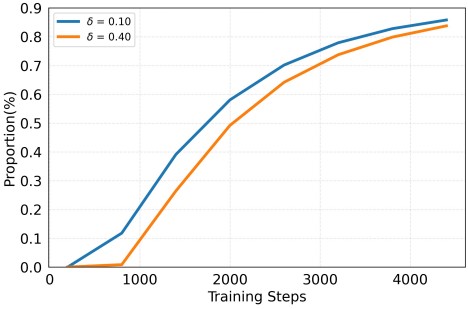

Figure 1: Variation of masked samples ratio on Clinc_oos.

In theory, the model with the training objective of AS-Softmax will gradually remove easy samples during training. As shown in Figure 1, experimental result shows taht the effective sample size in a batch of AS-Softmax is much smaller than that of softmax. Such few effective samples in a batch would waste computing resources such as the GPU memory. In addition, the hyper-parameters of the training optimizer such as the learning rate do not well match the training process in softmax. Therefore, we accumulate the training steps of AS-Softmax to make the effective sample size approach that of softmax. Specifically, we propose an adaptive gradient accumulation method according to the masked samples ratio:

$$steps_{accum} = \lambda * \frac{N_{all}}{N_{all} - N_{masked}}, \tag{19}$$

where $\lambda$ is a hyper-parameter controlling the accelerate magnitude and $N_{all}/N_{masked}$ denotes the number of all/masked samples. $\lambda$ is necessary as $steps_{accum}$ is a prediction of the successive batches.

In addition, we attach the following restrictions to the accelerate strategy to guarantee its effectiveness: i). making sure $steps_{accum}$ is increasing monotonically; ii). ensuring the difference between two adjacent $steps_{accum}$ does not exceed 1; and iii). setting an upper limitation to $step_{accum}$. For convenience, we call the acceleration algorithm of AS-Softmax as AS-Speed.

## 4 EXPERIMENTS

### 4.1 DATASETS AND EVALUATION METRICS

To evaluate our approach extensively, we conduct experiments on 6 different text classification datasets, including SST5 (Socher et al., 2013), Clinc_oos (Larson et al., 2019), Conll2003 (Sang & De Meulder, 2003), SIGHAN2015 (Tseng et al., 2015), Eurlex (Chalkidis et al., 2019) and WOS-46985 (Kowsari et al., 2017). These datasets belong to multi-class classification, token classification and multi-label classification tasks respectively. We separately adopt following evaluation metrics: accuracy, f1, macro/micro-f1 to measure multi-class classification, token classification and multi-label classification tasks. The basic information of the datasets and evaluation metrics is listed in Table 1. See Appendix A.2 for more details.

| Dataset | Task | # Train/Dev/Test | # classes | Model | Metrics |
|---------|------|------------------|-----------|-------|---------|
| SST5 | multi-class classification | 8544/1101/2210 | 5 | BERT | accuracy |
| Clinc_oos | | 15250/3100/5500 | 151 | BERT | accuracy |
| Conll2003 | token classification | 14042/3251/3454 | 9 | BERT | f1 |
| SIGHAN2015 | | 277786/1100/1100 | 5201 | Roberta | f1 |
| Eurlex | multi-label classification | 55000/5000/5000 | 127 | BERT | macro/micro-f1 |
| WOS-46985 | | 30070/7518/9397 | 141 | BERT | macro/micro-f1 |

Table 1: Statistics of experimental datasets.

### 4.2 BASELINES

We implement 7 baselines including **Softmax** and its six variants. We first compare our method with the temperature-based softmax (**T-Softmax**) which has hyper-parameter. Then, for sparse based algorithms, we choose **Sparse-Softmax** (Sun et al., 2021), **Sparsemax** (Martins & Astudillo, 2016) and **Entmax** (Peters et al., 2019) as our comparison methods. From the perspective of the reachable training objective, we introduce **Label-Smoothing** (Szegedy et al., 2016) method for comparison. In addition, $\delta$ plays the role of margin in AS-Softmax. Thus we compare AS-Softmax with the additive margin softmax (**AM-Softmax** (Wang et al., 2018c)). Most shared hyper-parameters refers to examples[2], while the specific hyper-parameters of each model are tuned on the validation set. Details of these models and their hyper-parameter settings are shown in Appendix A.3.

### 4.3 SETTINGS

Most experiments are conducted based on huggingface's pytorch implementation of transformers.The base pretrained models (Devlin et al., 2018; Liu et al., 2019) are initialized by BERT-base-cased and Roberta-wwm-ext respectively. We utilize AdamW (Loshchilov & Hutter, 2017) as the optimizer. Due to the little change of AS-Softmax to the original softmax code, it can be reproduced easily. Our experiments are mainly carried out on RTX A5000 with a memory size of 24G except that the experiment related to SIGHAN2015 is conducted on V100 with 32G memory. For more specific parameter settings of the tasks we used, please refer to the Appendix A.4.

---

[2]https://github.com/huggingface/transformers/tree/main/examples

| | SST5 accuracy | Clinc_oos accuracy | Conll2003 f1 | SIGHAN2015 f1 | Eurlex macro-f1 | micro-f1 | WOS-46985 macro-f1 | micro-f1 |
|---|---|---|---|---|---|---|---|---|
| Softmax | 51.90 | 88.60 | 90.56 | 70.80 | 46.87 | 68.73 | 80.40 | 86.00 |
| T-Softmax | **53.12** | 88.65 | 90.63 | 70.80 | 46.87 | 68.73 | 80.76 | 86.36 |
| Sparse-Softmax | 52.99 | 89.02 | 90.93 | 71.40 | 47.77 | **68.99** | 80.94 | 86.65 |
| Sparsemax | 51.49 | 88.55 | 90.57 | 71.49 | - | - | - | - |
| Entmax | 51.90 | 88.33 | 90.62 | **72.81** | - | - | - | - |
| Label-Smoothing | 52.85 | 88.64 | 90.64 | 68.10 | - | - | - | - |
| AM-Softmax | 52.17 | 89.05 | 90.66 | 68.57 | - | - | - | - |
| AS-Softmax | **53.12** | **89.07** | **91.03** | **72.81** | **48.30** | **68.99** | 80.94 | 86.39 |
| AS-Speed | 52.53 | 88.78 | 90.58 | 71.40 | 48.14 | 68.70 | **81.07** | **86.71** |

Table 2: Comparison of experimental results on various text datasets with different baselines ("-" means not available since their papers or codes do not provide the corresponding method for multi-label classification tasks).

| | SST5 | Clinc_oos | Conll2003 | SIGHAN2015 | Eurlex | WOS-46985 |
|---|---|---|---|---|---|---|
| Softmax | 288 | 533 | 625 | 10905 | 5944 | 5644 |
| Sparse-Softmax | -7% | +2% | -1% | -2% | -1% | -1% |
| Sparsemax | -2% | 0 | 0 | +8% | - | - |
| Entmax | 0 | 0 | -2% | +10% | - | - |
| Label-Smoothing | 0 | +1% | -3% | -4% | - | - |
| AM-Softmax | 0 | +1% | 0 | -3% | - | - |
| AS-Softmax | -1% | -1% | -1% | -2% | -1% | -2% |
| AS-Speed | **-22%** | **-15%** | **-21%** | **-7%** | **-13%** | **-5%** |

Table 3: Training time of different methods (in seconds). The training time of T-Softmax is not listed because it is the same as that of Softmax in theory.

## 4.4 MAIN RESULTS

The performance and training time on experimental tasks are presented separately in Table 2 and Table 3. Overall, the proposed AS-Softmax and AS-Speed have achieved improvements in terms of classification performance and efficiency in almost all the tasks.

From Table 2, we can observe that AS-Softmax consistently outperforms other methods. For multi-class classification tasks and token classification tasks, we find that AS-Softmax could bring more performance enhancement in the case of difficult tasks than simple tasks. It is worth noting that the f1 score of AS-Softmax on SIGHAN2015 is about 2 points higher than that of softmax. In terms of multi-label classification tasks, there is a slight difference between the training objective of AS-Softmax and the test goal. In testing, we will choose the classes with scores greater than 0 as the prediction results. However, in training period, AS-Softmax only encourages the probabilities of non-target classes to be lower than the minimum probabilities of target classes by a specific margin $\delta$. Despite the impact of this gap, the results of AS-Softmax also surpass softmax results on such datasets. More details will be analyzed in following sections.

Table 3 shows that all compared algorithms except AS-Speed perform fairly in terms of the training time. Notably, when it comes to high-dimensional output classes (i.e. SIGHAN2015), it may cost more time for those methods which requires $O(nlogn)$ computation complexity, such as Entmax and Sparsemax. Compared with softmax, AS-Speed could increase the training speed by about $1.2\times$ on the premise of maintaining the classification accuracy. To some extent, the acceleration effect depends on the number of classification categories. AS-Speed has achieved greater speed improvement on SST5 and Conll2003 which have no more than 10 output classes. Meanwhile, AS-Softmax also has more than $1.1\times$ acceleration performance over softmax although there are 150+ output classes such as Clinc_oos. For SIGHAN2015 and WOS-46985, AS-Speed has no significant effect. One important reason is that there are too many output classes and the proportion of masked samples is unstable, leading to the maximum accumulated steps can only be 2.

In additional, we conduct experiment on some classification tasks of GLUE benchmark Wang et al. (2018a). The result are presented in Appendix A.6.

## 4.5 RESULT ANALYSIS

**Impact of** $\delta$   The $\delta$ value can significantly affect the accuracy of the classifier. If $\delta$ is too large, AS-Softmax would not mask such easy samples, making the model to overlearn a lot of useless information. In contrast, AS-Softmax would cast away some potential useful knowledge when $\delta$ is rather small, which could result in insufficient learning. In our experiments, we attempt different values of $\delta$ ranging from 0.05 to 0.5 and choose the final value which achieves the highest classification performance on the development set. According to such experiments, we suggest setting $\delta$ to a moderate value 0.3. Then we investigate the influence of $\delta$ in the case of relative easy and hard tasks. As Figure 2 shown, the predictions on SST5 represents a difficult situation while the result on Conll2003 denotes a simple case. It can be seen that the model performs best on SST5 when the value of $\delta$ is 0.2 (relative smaller). By contrast, a slightly larger $\delta$ is more helpful to its classification effect on Conll2003. In conclusion, we believe that it could be better to set $\delta$ to a relative larger value for simple tasks and set $\delta$ to a smaller value for difficult tasks.

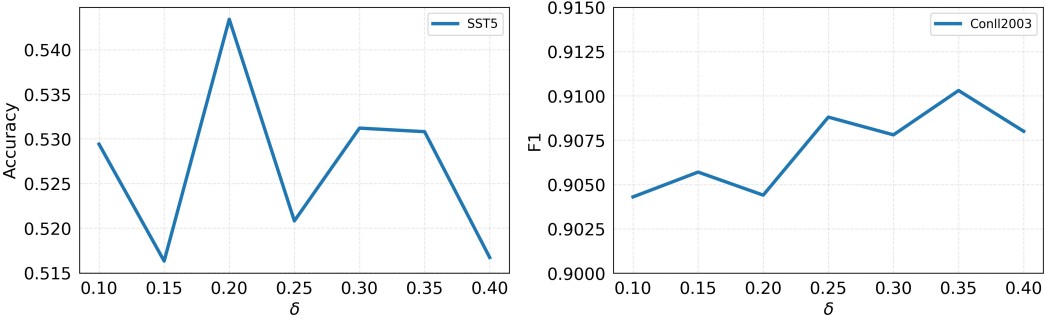

Figure 2: Accuracy/F1 metrics on SST5 and Conll2003 with different $\delta$.

|  | SST5 | Clinc_oos | Conll2003 | SIGHAN2015 | Eurlex | WOS-46985 |
|---|---|---|---|---|---|---|
| Softmax | 0.038 | -0.943 | -0.975 | -0.947 | -0.337 | -0.902 |
| Sparse-Softmax | 0.088 | -0.914 | -0.981 | -0.907 | -0.114 | -0.901 |
| Sparsemax | 0.050 | -0.976 | -0.968 | -0.681 | - | - |
| Entmax | 0.080 | **-0.986** | -0.983 | -0.617 | - | - |
| Label-Smoothing | -0.087 | -0.952 | **-0.998** | **-0.989** | - | - |
| AM-Softmax | 0.057 | -0.915 | -0.974 | -0.982 | - | - |
| AS-Softmax | **-0.952** | -0.949 | -0.996 | -0.986 | **-0.909** | **-0.904** |

Table 4: Results of Pearson correlation coefficient between loss and accuracy.

**Correlation between Loss and Classification Performance**   To verify that our training objective can alleviate the train-test goal mismatching problem, we introduce the Pearson correlation coefficient which measures the relationship between the loss and the classification ability in validation. The correlation coefficient is close to - 1 or 0 indicating the loss is linear or has no significant correlation with the classification performance. If the correlation coefficient is close to 1, it demonstrates that the learning direction of the model is obviously opposite. In theory, the final classification performance is inversely correlated with the loss. Table 4 illustrates that the correlation between loss of all methods and their classification performance is linear on datasets with high identical accuracy such as Conll2003 and SIGHAN2015. However, when it comes to hard datasets (i.e. SST5 and Eurlex), the correlation coefficients of compared baselines are not good enough and some correlations could be even the opposite. By comparison, the loss of AS-Softmax and its classification performance is highly relevant across whole experimental datasets, which verified the design of AS-Softmax training objective.

To intensively explore the correlation between the loss and classification performance, we record the loss and accuracy on development set of SST5. The left/right subfigure of Figure 3 represents its classification performance and normalized loss, respectively. It can be seen that other models tend to overfit as their accuracy is still fluctuating while the loss has turned to increase, after 1000+ training steps. By contrast, the loss of AS-Softmax does not increase significantly at the end of training

process, which also reveals that it could moderate the overfitting problem. To verify the discarded samples are easy samples we expected, we do more experiment. See details in Appendix A.5.

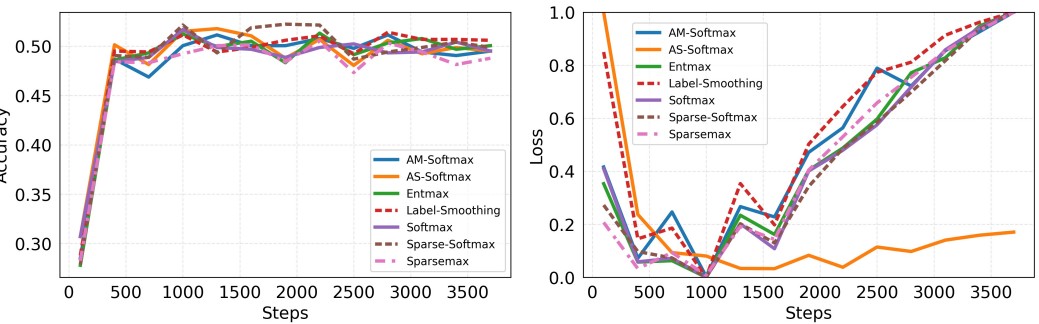

Figure 3: Comparison of development performance on SST5.

**Probability Margin of Positive and Negative Classes** The AS-Softmax training goal is calculated according to target probability $p_t$ and non-target probability $p_{i \neq t}$. To check this objective, we design a metric $p_{margin} = p_t - p_{neg}^{max}$, where $p_{neg}^{max}$ indicates the largest probability in non-target classes. $p_{margin}$ is greater or less than 0 denoting that the classification result is correct or wrong. Then we draw the distribution diagram of $p_{margin}$ on SST5 in Figure 4, where $\delta$ in AS-Softmax is set to 0.3. Generally speaking, Softmax result is widely distributed while the distribution of AS-Softmax result is very concentrated. On one hand, when $p_{margin}$ is less than 0, AS-Softmax pushes $p_{margin}$ close to 0. It indicates that even if the true label is not identified correctly, its top k candidates in the prediction result still have some reference value. By comparison, there are many strong negative classes in the softmax result, which makes the result completely unavailable. On the other hand, when $p_{margin}$ is greater than 0, the overall distribution of $p_{margin}$ is squeezed to be lower than or close to $\delta$. It suggests that AS-Softmax could prevent the model from being optimized towards overlearning easy samples and the model could pay more attention to those hard samples. To conclude, the probability margin distribution is in accordance with the proposed training objective which encourages the target class to exceed other non-target classes by a specific margin $\delta$.

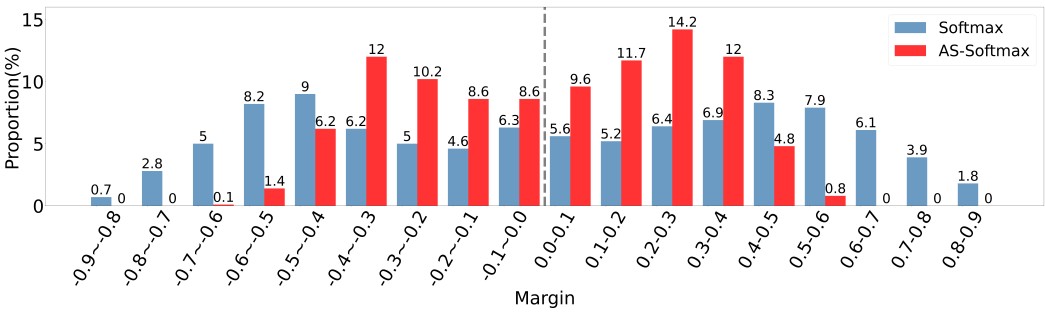

Figure 4: Distribution diagram of $p_{margin}$ on the SST5 test set with $\delta = 0.3$. We add a dotted line to stand for $p_{margin} = 0$, which splits the wrongly (left) and correctly (right) predicted cases.

## 5 CONCLUSION

In this paper, we propose a simple yet effective softmax variant, namely adaptive sparse softmax (AS-Softmax), which discards easy training samples at the aim of a more reasonable and test-matching learning objective. We further develop an adaptive gradient accumulation strategy based on the masked sample ratio to accelerate the training process of AS-Softmax. Experimental results on 6 text classification datasets show that AS-Softmax consistently surpasses the original softmax and its variants. We believe our work can be extended in many aspects. On the one hand, instead of the current fixed value of $\delta$, we plan to investigate an improvement strategy which decides $\delta$ adaptively. On the other hand, we are curious about the performance of AS-Softmax on the generation tasks and other input modals such as images or audio.

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

## A  APPENDIX

### A.1  AS-VARIANT

AS-Softmax is to mask the weak negative class features after obtaining the probabilities of classes. One feature of AS-Softmax is $\delta \in [0, 1]$. It is convenient to find the optimal $\delta$ value of AS-Softmax. We also examine a simple training criterion. That is, apply $\delta'$ to the final output $o$ of the neural classifier. Similar to the principle of AS-Softmax, $\delta'$ is a margin used to open the gap between the scores of target categories and that of other categories. Nevertheless, $o$ can take any value, namely no upper bound of $\delta'$, which makes it hard to find the optimal value. We call the above method AS-variant. We carry out the experiment of AS-variant on SST5.

In the Figure 5, the gray dotted line represents the prediction results of original softmax while the experimental results of AS-Softmax are indicated by the red dotted line. The blue one represents

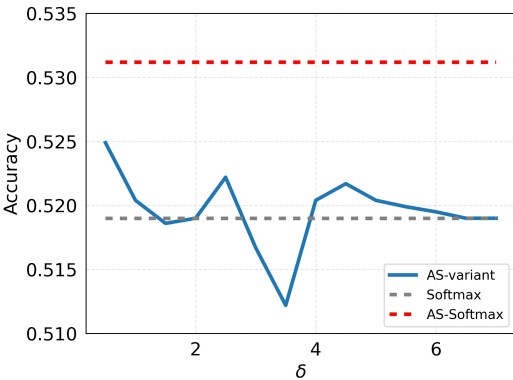

Figure 5: Accuracy of AS-variant with different $\delta'$ on SST5 test set.

that the results of AS-variant change with $\delta'$. We take the value of $\delta'$ from 0 to 7. As can be seen from this figure, when $\delta'$ is greater than 6, AS-variant is equivalent to softmax. It can be seen that the prediction results fluctuate up and down around the baseline. Therefore, we can not determine where to obtain the optimal $\delta'$ under this algorithm as easily as AS-Softmax. Moreover, the results are not as good as AS-Softmax.

## A.2 DATASETS AND METRICS

In this section, we describe the experimental datasets and metrics in details.

**SST5.** This dataset (Socher et al., 2013) is a sentence level classification dataset for sentiment analysis. It contains 5 output classes (from very negative to very positive). It is made up of 8544 train, 1101 validation, 2200 test samples. We adopts pretrained language model BERT initialized with bert-base-cased parameters. The evaluation metric for this dataset is accuracy.

**Clinc_oos.** This dataset (Larson et al., 2019) is for intent classification, comprising of 151 classes over 10 domains. Its train, validation, test dataset contains 15250, 3100, 5500 instances. We use BERT as our backbone. Accuracy is also used as evaluation metrics.

**Conll2003.** This dataset (Sang & De Meulder, 2003) is for named entity recognition (NER) task. It consists of four types of name entities: persons, locations, organizations and others. There are nine categories in its output. The sizes of train, validation, test dataset are 14042, 3251, 3454, respectively. We use f1 score to measure this dataset.

**SIGHAN2015.** This dataset (Tseng et al., 2015) is designed for the Chinese Spelling Check (CSC) task. Following previous works (Li et al., 2021; Cheng et al., 2020), we formula CSC as a token classification task whose candidates are tokens in vocab. The training dataset with the size of 277786 is comprising of the automatic generated dataset (Wang et al., 2018b) and all SIGHAN training sets (Wu et al., 2013; Yu et al., 2014; Tseng et al., 2015). The test set is the SIGHAN2015 test (Tseng et al., 2015) with a size of 1100. Considering the tokens we focus are Chinese characters, we cut the original vocab according to the training data. Finally there are 5201 characters left. We follow previous works (Cheng et al., 2020; Li et al., 2021) and choose the metric of sentence level correction-f1 which could more strictly represent the classifier competence.

**Eurlex.** This dataset (Chalkidis et al., 2021) is for legal multi-label classification task. All EU laws are annotated by the EU Publications Office with multiple concepts from the eurovoc dictionary, a multilingual dictionary maintained by the Publications Office. Given a document, the task is to predict its EuroVoc labels (concepts). It contains 55000, 5000, 5000 instances for training, validation and test. There are 127 labels in this dataset, and each sample has multiple labels. We use BERT as our base model and macro-f1/micro-f1 to evaluate the classifier performance.

**WOS-46985.** We choose WOS-46985 (Kowsari et al., 2017) as the dataset of multi-label classification task. Obviously, the total size of this dataset is 46985. Following Wang et al. (2022), we split the dataset into training set, validation set and test set according to 6.4:1.6:2 split the dataset into

training set and test set. The dataset consists of three types of labels: target label, parent label and child label. We combined the parent label with the child label as the output classes following Wang et al. (2022). Finally, the number of output classes in this dataset is 141. Notably, there are two corresponding labels for each sample and we only select the largest two classes as the result in test. We also use BERT as our backbone and macro-f1/micro-f1 as the evaluation metrics.

## A.3 BASELINES

The comparison algorithms used in our experiment are described in details here.

**T-Softmax.** T-Softmax is a simple variant of introducing the temperature parameter $\tau$ into Softmax. Divide the model output by $\tau$. The larger $\tau$ is, the smoother the result is. The smaller the corresponding $\tau$ is, the sharper the probability distribution is. The prediction score of a class $p_i$ in T-Softmax is computed as follows:

$$p_i = \text{T-Softmax}(o_i) = \frac{e^{\frac{o_i}{\tau}}}{\sum_{j=1}^{n} e^{\frac{o_j}{\tau}}}. \tag{20}$$

**Label-Smoothing.** The authors of Label-Smoothing (Szegedy et al., 2016) argue that these non-target classes need to be given a probability greater than zero so that the model can fully learn all classes' features. Considering that the dataset is not completely correct, the original softmax loss function is even less applicable. Label-Smoothing has a parameter $\epsilon$. In the softmax loss function, the calculated weight of non-target classes is 0. However, in the label smoothing algorithm, assuming that a dataset has $n$ labels, the weight of a non-target class is $\frac{\epsilon}{n-1}$. The weight of the target class is $1 - \epsilon$. $\epsilon$ is a hyper-parameter of label smoothing.

**Sparse-Softmax.** Sparse-Softmax (Sun et al., 2021) means that softmax results are sparse. Sparse is effective because it avoids overlearning of the model. This algorithm only preserves the features of the top $k$ classes, and the remaining features are directly assigned to 0 for mask. Obviously, $k$ is a hyper-parameter.

**AM-Softmax** Inspired by algorithms such as large-margin, additive margin softmax (AM-Softmax) (Wang et al., 2018c) aims to minimize the intra-class variation. And the loss function and back propagation process of large-margin softmax are too complicated due to the introduction of angle. AM-Softmax not only simplifies the calculation process, but also speeds up the training speed. Its loss function is as follows:

$$\mathcal{L} = -\log \frac{e^{s \cdot (o_i - m)}}{e^{s \cdot (o_i - m)} + \sum_{j=1, j \neq i}^{n} e^{s \cdot o_j}}. \tag{21}$$

It has two parameters, $s$ and margin $m$. $m$ is used to increase the margin between categories. $s$ has the effect of acceleration. When $s$ or $m$ is too large, NAN may occur in the value of the loss function. Therefore, $s$ and $m$ vary greatly with different datasets. This is a process that needs to be adjusted slowly.

**Sparsemax.** Sparsemax (Martins & Astudillo, 2016) returns sparse posterior distributions by assigning zero probability to its output variables with small scores.

**Entmax.** Entmax algorithm (Peters et al., 2019) tends to produce sparse probability distributions, yielding a function family continuously interpolating between softmax and Sparsemax. There are two algorithms including a method based on sorting and a bisection-based method. We do a comparative experiment with the 1.5-entmax proposed by author (Peters et al., 2019). It is based on sorting when $\alpha$ is equal to 1.5 and more accurate and less costly.

## A.4 EXPERIMENTAL DETAILS

In our experiment, we additionally adopt special warm-up strategy on Clinc_oos and Conll2003. At the beginning of training period, AS-Softmax may discard some potential useful samples due to the poor classifier competence. Therefore, we keep $\delta$ equal to 1 in the first $r$ percent of training steps. For AS-Speed algorithm, there are two more hyper-parameters: $\lambda$ and $s$. $\lambda$ is a hyper-parameter which controls the accelerate magnitude while $s$ denotes maximum accumulated steps. In addition,

we implement Entmax, Sparsemax and AM-Softmax according to the source code[34]. We record the general and specific hyper-parameters respectively in Table 5 and Table 6.

| | SST5 | Clinc_oos | Conll2003 | SIGHAN2015 | Eurlex | WOS-46985 |
|---|---|---|---|---|---|---|
| epoch | 7 | 10 | 20 | 15 | 20 | 10 |
| batch size | 16 | 32 | 32 | 128 | 16 | 12 |
| learning rate | 2e-5 | 2e-5 | 2e-5 | 5e-5 | 3e-5 | 3e-5 |

Table 5: Information of universal hyper-parameters.

| | SST5 | Clinc_oos | Conll2003 | SIGHAN2015 | Eurlex | WOS-46985 |
|---|---|---|---|---|---|---|
| T-Softmax ($\tau$) | 5 | 0.5 | 0.5 | 0.5 | 1 | 0.5 |
| Sparse-Softmax ($k$) | 4 | 10 | 5 | 20 | 50 | 100 |
| Entmax ($\alpha$) | 1.5 | 1.5 | 1.5 | 1.5 | - | - |
| Label-Smoothing ($\epsilon$) | 0.30 | 0.20 | 0.10 | 0.10 | - | - |
| AM-Softmax ($s/m$) | 10/0.30 | 8/0.35 | 5/0.30 | 1/0.10 | - | - |
| AS-Softmax ($\delta/r$) | 0.30/0.00 | 0.30/0.15 | 0.35/0.25 | 0.30/0.00 | 0.05/0.00 | 0.35/0.00 |
| AS-Speed ($\lambda/s$) | 1.5/4 | 0.5/5 | 0.5/5 | 0.5/2 | 0.5/2 | 1.0/2 |

Table 6: Parameter setting of algorithm. $s$ after AM-Softmax means scale. $s$ after AS-Speed means max accumulation steps.

## A.5 SAMPLE ANALYSIS

To verify the "easy" samples are not selected by chances, we construct the datasets with hard samples left at the end of the AS-Softmax training period on SST5 (approximately 11% of all) and the same number of random selected samples. Then, we train the classifier with softmax and the result is shown in the Table 7. The performance illustrates that these hard samples are more difficult than the random selected samples for the model to learn.

| | SST5 |
|---|---|
| Random samples | 45.52 |
| Hard samples | 38.64 |

Table 7: Result of standard softmax method on random sample dataset and hard sample dataset.

## A.6 RESULTS ON GLUE DATASET

We further evaluate our proposed AS-Softmax on the classification tasks of GLUE benchmark (Wang et al., 2018a), including MNLI, MRPC, QNLI, QQP, RTE, SST-2 and CoLA. The shared hyper-parameters are set following Liu et al. (2019) while the margin $\delta$ in AS-Softmax is adjusted in [0.05, 0.35] for searching. Table 8 presents the experimental result. It can be seen that the result of AS-Softmax surpasses that of Softmax, which verifies its effectiveness.

| | MNLI:3 | MRPC:2 | QNLI:3 | QQP:2 | RTE:3 | SST-2:2 | CoLA:2 | Avg |
|---|---|---|---|---|---|---|---|---|
| Softmax | 84.18 | 89.41 | 90.49 | **91.00** | 68.23 | 92.54 | 62.44 | 82.61 |
| AS-Softmax | **84.19** | **90.72** | **91.12** | 90.95 | **69.31** | **93.23** | **64.55** | **83.44** |

Table 8: Result of GLUE benchmark (better results are in **bold**), with the number of output class attached behind the task name.

---

[3] https://github.com/deep-spin/entmax
[4] https://github.com/happynear/AMSoftmax

