# OpenReview forum: "Adaptive Sparse Softmax: An Effective and Efficient Softmax Variant for Text Classification"
_ICLR.cc/2023/Conference — Submitted to ICLR 2023_

### Official Review · Reviewer_UMbA · 2022-10-17

**Confidence:** 3
**Clarity, Quality, Novelty And Reproducibility:** Please see the summary section.
**Correctness:** 2
**Technical Novelty And Significance:** 2
**Empirical Novelty And Significance:** 2
**Recommendation:** 3

**Strength And Weaknesses:**

Reasons to accept:

The idea is intuitive and is easy to implement.

Reasons to reject:

- I found the arguments discussed in the paper weak and incomplete.
  - Example 1: Page 1, second paragraph, the first three lines, the authors state: “the weights of all the classes have to be updated endlessly, which wastes a lot of time and leads to overfitting”.

    In my opinion, this is an incorrect depiction of the issue. The classes in neural nets share their weights. The authors show that their model speeds up the training. The cause of this acceleration is not omitting the classes, but it is dropping the easy examples.
  - Example 2: Page 1, second paragraph, Lines 4-5, the authors state: “the training goal of softmax with the cross entropy loss is to make the target score approach to 1, while in test we expect the target score could be superior to scores of other classes.”

    This is incorrect. The accurate output of softmax function has many applications in Bayesian learning, in active learning, in model explainability, etc.

- The authors have done a poor job in contrasting their work against other similar studies. In the related work section they simply list the references, without explicitly discussing their distinctions. They have done the same in Section 4.2. Given this, I would say their proposed model is a small extension of the “Sparse-Softmax”, and is not worth publishing as a full paper at a top conference.

- I believe the closest area of study to the proposed model is curriculum learning, which the authors have not discussed it (or aren’t aware of it) at all. No discussion, no related work, and no baselines on the subject.

- You would need a magnifier to see the improvements, they are not noticeable to publish.

- The writing is very poor, I won’t even bother giving examples, starting from the title (specifically the title in the pdf file) up to the end.

**Summary Of The Paper:**

The authors propose a weighting variant of the softmax function. The weights are binary and are a function of the difference between class probabilities subject to a threshold.

The authors evaluate their model in six datasets and show that it achieves a small improvement over other variants. They also show that their model is faster in training and the speed-up is noticeable in some cases.

**Summary Of The Review:**

The idea is intuitive and easy to implement, but the arguments in the paper are weak, the related work section is insufficient, the authors have missed to consider curriculum learning, the improvements are not noticeable, and finally the presentation is horrible.

---

### Official Review · Reviewer_bnPp · 2022-10-18

**Confidence:** 3
**Correctness:** 3
**Technical Novelty And Significance:** 2
**Empirical Novelty And Significance:** 2
**Recommendation:** 5

**Clarity, Quality, Novelty And Reproducibility:**

**Clarity**. I believe that you could rewrite some parts of the paper for added clarity. For instance, in $\S3.2.2.$, you refer to a so-called “sigmoid based method” without references and  without explaining what that is. Although I know what you are referring to, I don’t think this is very clear. Also, the related work section is not very enlightening (see my comments above).

**Quality**. See strengths/weaknesses presented above.

**Novelty**. As introduced in $\S3.2$, adaptive sparse softmax (the proposed approach) is very similar in nature to sparse-softmax (Sun et al., 2021), which preserves a fixed top-k number of classes during training (this is not the case for adaptive sparse softmax). However, contrary to the sparse-softmax loss, the loss of adaptive sparse softmax is not always larger than zero. That being said, I still think the novelty is limited.

**Reproducibility**. Code is provided as supplementary material and there is a note saying it will be released in the final version.


**Strength And Weaknesses:**

Strengths:

* The motivation is clear.
* AS-softmax seems to be an interesting  and easy-to-implement way to address some of the problems with softmax + cross entropy models.

Weaknesses:

* The related work section is too short and needs to be reviewed. Although you are already citing several works that attempt to solve some of the problems you are trying to tackle, the discussion is very superficial, making it hard to understand the difference between each of the baselines you consider (e.g., sparse-softmax vs sparsemax vs entmax). In particular, sparse-softmax should be clearly discussed here, since your proposal is highly influenced by this method. Crucially, some of the functions that you are using as a baseline in $\S4$, are not even included in the discussion in $\S2$. Although I understand that they are presented in App. A.3 (some of them are not very detailed), I think the paper could benefit a lot from this discussion at an earlier stage. Besides, you used the term “softmax loss” several times, which does not sound accurate to me – softmax is an activation function, not a loss.
* The experimental results are not very convincing. Although AS-Softmax tends to perform generally better than the baselines, there is a drop in the performance when using the AS-Speed. Can you elaborate on why you think this happens?
* I found some inconsistencies/typos (listed below), please revise them.

Other questions:
* Can you please provide further information on how AS-Softmax works at test time? I don’t think the current version of the paper discusses that.
* Can you please explain how you tuned the hyperparameters for all the methods presented in Table 2? I may have missed something but I could not find this information for all the methods you experimented with.

Minor comments:

* Please update the title
* “Then the model could focus on learning to distinguish the target class from its strong opponents, which is also the great challenge in test.” -> I don’t think this sentence is clear.
* Typo (*sparse*) in “we propose the Adaptive Spare softmax (AS-Softmax)”
* Typo (remove whitespace before the comma) in “will be dropped from back propagation gradually ,”
* Typo (*preserve*) in “Jean et al. (2014); Rawat et al. (2019) and Blanc & Rendle (2018) preserves”
* Not using the right citation format in “The features learned by softmax loss have intrinsic angular distribution Liu et al. (2017).”
* Typo (*a*) in “We have made it as an standard”
* Typo in “of the original the exponential function”
* Typo (*that*) in “As shown in Figure 1, experimental result shows taht”
* What do you mean by “Most shared hyper-parameters refers to examples”?
* Typo in “In additional, we conduct experiment on some (...) The result are presented”.
* Typo (*shows*) in “As Figure 2 shown”



**Summary Of The Paper:**

This paper provides an alternative to the softmax + cross entropy loss widely used in text classification tasks. The proposed method, named adaptive sparse softmax (AS-Softmax), is inspired by the sparse-softmax (Sun et al., 2021) and changes the training objective by including a binary term that, when multiplied by a model’s output, excludes from the loss computation the classes that already satisfy a given margin (using an hyperparameter $\lambda$). They further propose an adaptive gradient accumulation strategy to make the training procedure faster. Finally, they perform experiments on multi-class, multi-label, and token classification tasks, showing slightly better results over the baselines.

**Summary Of The Review:**

Although the work is well motivated and the problem they try to address is relevant, I don’t think the paper is ready for publication. The main reasons are explained in the Strengths/Weaknesses section.

---

### Official Review · Reviewer_EExx · 2022-10-23

**Confidence:** 4
**Correctness:** 3
**Technical Novelty And Significance:** 2
**Empirical Novelty And Significance:** 2
**Recommendation:** 6

**Clarity, Quality, Novelty And Reproducibility:**

I have following questions for the paper and I hope authors could answer it thoroughly.

1. I don't quite get the gist of your motivating example : " in the training period, since the target score in Case B is much
higher than that in Case A, the corresponding cross entropy loss would make the model biased
towards improving Case A, although Case B is a hard sample in practice"

To me it's not "much higher" for Case B, the loss I calculated is about "1.35" and for case A is "1.41". I don't see the reason the training will completely neglect case B but only work on case A. What if after say 3 iterations, Case  B has the correct prediction but still has smaller loss to case A. There will be no problem for the example then.

2. I think you mentioned 2 problems in the paper. First is overfitting due to unable to get perfect loss. Second is training efficiency due to residual probabilities. In the case of overfitting, that means using AS-softmax could easily make training loss to 0. did you provide any justification to this claim ? Also, if the overfitting is the problem. People could just use early stopping. That kinds confused me, how many epochs you used to train other baseline methods such as softmax/ t-softmax/ Label-Smoothing ? Say if you use 20 for them. Then according to your claim, it's possible for use to observe a much better performance of these methods at iteration 18 or earlier. Can you justify this part by providing more details on how you use baselines?


3. Following point 2, if overfitting is the issue. What we want to see is a result plot of training loss versus validation loss plots instead of a simple table (such as table 2) to summarize the accuracy results. I believe this is another way to answer 2. Providing a complete plot of accuracy/losses versus iterations plot for each method to make us understand what's really going on.

4. In Section 3.1, you mentioned "Sun et al. (2021) proved that in order to make sure the loss L can be reduced to log 2, the output should satisfy the following inequality". But why we need to reduce it to log 2?

5. I am not sure what's the purpose of your 3.2.1 ALGORITHM DISCUSSION. You are comparing the requirement to eq(4) but what's the guarantee you can get? In (4) but guarantee is you can the loss L can be reduced to log 2 ( despite I don't know why we need to have so) but here in 3.2.1 what's the corresponding guarantee?

6.  To me, \delta seems to be a hyper-parameter and it's fairly critical. Initially, I was wondering how the experiments is done and I found out in the Appendix " At the beginning of training period, AS-Softmax may discard some potential useful samples due to the
poor classifier competence. Therefore, we keep δ equal to 1 in the first r percent of training steps."

Then I got 2 follow-up questions.
1) How is r determined ? I guess it's tuned. If that's the case, I really think the time to tune r should be counted as training time so overall the training time is not 1.2x faster but rather slower. If it's not tuned, I hope authors can illustrate a rigorous plan to find such r.

2) I also think temperature softmax can adopt similar strategies as well. t hyperparmeter is designed to function as what this paper want to achieve too : too diminish residual values. If you could change \delta along the experiment, it's also important to change temperature along the experiments too. Please also fine-tune the t-softmax hardly too and discuss what's the result.

7. I don't quite see how it is likely that the correlation between loss and accuracy is nearly zero but accuracy is still very high in SST5. AS-SOFTMAX has high correlation but somehow accuracy is not much better. I think that's even a more worth studying research problem if it's true. I didn't find authors addressed this in depth or did you write it anywhere? I believe again we need a more thorough loss/accuracy per iteration plot to see what's going wrong.









**Strength And Weaknesses:**

Strength:
1. An interesting problem.
2. Reasonable method to work on it.
3. Moderately enough experiments.

Weakness:
1. Presentation is confusing. Motivating example is not demonstrating the point.
2. Not enough experiments to justify the articulated deficiencies is the root cause.


**Summary Of The Paper:**

Authors believe there are problems with naive softmax function used in ML training. The design of softmax will lead to 2 potential problems: overfitting and mismatch between softmax loss and prediction goal. They proposed AS-SOFTMAX to address so. Authors addressed how these 2 problems could affect the final performance. But a more rigorous analysis is not presented. Experimental results found some interesting problems such as low correlation between accuracy and loss for certain datasets.

**Summary Of The Review:**

Overall, I think this research question is interesting and the proposed method is certainly a doable solution. However, I do think the analysis is not in depth enough to convince me the efficacy. To encourage authors to justify their method, I am more leaning toward acceptance at this point of time, and hope authors resolve all of my confusions mentioned above.

---

### Official Review · Reviewer_FqyY · 2022-10-30

**Confidence:** 3
**Correctness:** 3
**Technical Novelty And Significance:** 2
**Empirical Novelty And Significance:** 2
**Recommendation:** 5

**Clarity, Quality, Novelty And Reproducibility:**

- Clarity: the writing is clear and easy to follow
- Quality and Novelty: the proposed method seems somehow heuristic which lacks theoretical justification. The novelty is also limited, which seems to be a mild extension of Sparse-Softmax (Sun et al. AAAI 2022) paper.
- Reproducibility: no code provided, but the method seems rather simple to implement and reproduce

**Strength And Weaknesses:**

**Strength**
- The paper writing is clear to follow

**Weakness**
- The proposed method lack theoretical justification. Several theoretical analysis are missing. For example, (1) How does it resolve the overfitting issue? Can you show AS-Softmax has better generalization error? (2) What metric is AS-Softmax optimizing? Is it a consist estimator?
- AS-Softmax introduced additional hyper-parameter delta, which requires extra hyper-parameter tuning
- The experiment results are somehow weak with room to improve. For example, (1) What's the performance on the multi-label problem with extreme large output space, such as Wiki-500K and Amazon-3M? (2) What's the performance on image classification such as ImageNet-1K, which also has large number of classes?


**Summary Of The Paper:**

This paper argued that Softmax cross entropy loss function may lead to overfitting for large-output space problems. To resolve such issue, the author propose Adaptive Sparse Ssoftmax (AS-Softmax) that masked out a non-target logit when the target logit exceed the non-target logit by a specific margin. The author also extends AS-Softmax to the multi-label setting. Experiments are conducted on standard text classification datasets, where AS-Softmax demonstrated marginal improvement over other Softmax variants.

**Summary Of The Review:**

This paper has rather limited novelty, lacks theoretical justification (Weakness-1), and rather preliminary experiment results (Weakness-2). Hence, I am not inclined accepting this paper.

---

### Decision · Program_Chairs · 2023-01-20

**Decision:**

Reject

**Justification For Why Not Higher Score:**

The paper need a significant revision before publishing.

**Justification For Why Not Lower Score:**

N/A

**Metareview: Summary, Strengths And Weaknesses:**

This paper presents a study of adaptive sparse softmax that discards classes with smaller scores and focuses on distinguishing between the target class and its strong opponents.

Strengths:
+ The idea is interesting and can have a great potential impact as softmax is used in many applications

Weaknesses:
- Lack of theoretical justification for the proposed approach
- The motivation is not clear and some arguments in the paper are handwaving, confusing, and ambiguous.
- The experiment results are weak and do not fully support the claims. More analysis is needed.

There is no author rebuttal.



**Summary Of Ac-Reviewer Meeting:**

N/A